# Acute Effects of Sedentary Behavior on Ankle Torque Assessed with a Custom-Made Electronic Dynamometer

**DOI:** 10.3390/jcm11092474

**Published:** 2022-04-28

**Authors:** Iulia Iovanca Dragoi, Florina Georgeta Popescu, Teodor Petrita, Florin Alexa, Sorin Barac, Cosmina Ioana Bondor, Elena-Ana Pauncu, Frank L. Bowling, Neil D. Reeves, Mihai Ionac

**Affiliations:** 1Department of Vascular Surgery and Reconstructive Microsurgery, “Victor Babes” University of Medicine and Pharmacy, 2 Eftimie Murgu Square, 300041 Timisoara, Romania; contactfastfizioclinic@gmail.com (I.I.D.); sorin.barac@umft.ro (S.B.); frank.bowling@manchester.ac.uk (F.L.B.); mihai.ionac@umft.ro (M.I.); 2Discipline of Occupational Health, “Victor Babes” University of Medicine and Pharmacy, 2 Eftimie Murgu Square, 300041 Timisoara, Romania; medicinamuncii@umft.ro; 3Department of Communications, Politehnica University Timisoara, 2 Vasile Parvan, 300223 Timisoara, Romania; florin.alexa@upt.ro; 4Department of Medical Informatics and Biostatistics, University of Medicine and Pharmacy “Iuliu Hatieganu”, 8 Victor Babes, 400000 Cluj-Napoca, Romania; cbondor@umfcluj.ro; 5Department of Surgery & Translational Medicine, Faculty of Medical and Human Sciences, University of Manchester, Oxford Road, Manchester M13 9PL, UK; 6Research Centre for Musculoskeletal Science & Sports Medicine, Department of Life Sciences, Faculty of Science and Engineering, Manchester Metropolitan University, Oxford Road, Manchester M1 5GD, UK; n.reeves@mmu.ac.uk

**Keywords:** prolonged sitting, ankle torque, dynamometer, muscle strength, physical activity, sedentary behaviour

## Abstract

Inactivity negatively influences general health, and sedentary behaviour is known to impact the musculoskeletal system. The aim of the study was to assess the impact of time spent in active and sedentary behaviour on foot muscle strength. In this observational study, we compared the acute effects of one day of prolonged sitting and one day of low-to-moderate level of activity on ankle torque in one group of eight healthy participants. Peak ankle torque was measured using a portable custom-made electronic dynamometer. Three consecutive maximal voluntary isometric contractions for bilateral plantar flexor and dorsiflexor muscles were captured at different moments in time. The average peak torque significant statistically decreased at 6 h (*p* = 0.019) in both static and active behaviours, with a higher average peak torque in the active behaviour (*p* < 0.001). Age, gender, body mass index and average steps did not have any significant influence on the average value of maximal voluntary isometric contraction. The more time participants maintained either static or active behaviour, the less force was observed during ankle torque testation. The static behaviour represented by the sitting position was associated with a higher reduction in the average peak ankle torque during a maximal voluntary isometric contraction when compared to the active behaviour.

## 1. Introduction

Human physical activity and physiology of movement during daily activities, professional activities and sports in all age-related populations have gained increased scientist’s interest. Physical activity level influences general health, and the expansion of sedentary behaviour and its chronic complications are of great concern lately. Studying how sedentary behaviour influences human health across a lifespan might help increase physical and psychological wellbeing among various populations. 

Physical activity has been defined as movement produced by the action of the skeletal muscles and can be related to occupational, household, sports or any other activities. Physical fitness has been defined as planned, structured exercise and is different from physical activity. These two particular terms are often confounded. Physical fitness might be misused when defining physical activity [1].

When compared with the rest basal level, activity has been associated with higher energy expenditure, while the absence of activity has been defined as inactivity [1]. For a better thermology statement, the World Health Organization [2] and the National Institute of Health [3] have these definitions stated. Undertaking 7000–8000 steps/day is the border for physical activity to be categorised [4]. 

A sedentary lifestyle is considered when ≤5000 steps/day are undertaken, while ≈3500 steps/day is associated with an extremely low level of activity [5]. The same number of steps (just below 5000 steps/day) have been reported in the Framingham Heart Study [6]. 

Low-intensity physical activity (LIPA) and moderate to vigorous physical activity (MVPA) are the main two types of identified activity levels. A minimum of 150–300 min of weekly physical activity with moderate intensity, or 75–150 min of vigorous activity, or a combination of both types of activities is highly recommended [7]. By replacing the sedentary-spent time with light/moderate/heavy activities, a positive influence could be obtained on the human body’s main functions [8]. Increasing heavy physical activity for only five minutes can resemble a reduction of one hour of sedentary time [9]. 

Despite the well-documented general health benefits of MVPA [10], the adult population still shows a high amount of daily sedentary behaviour time [11]. Sedentary behaviour has shown negative effects, with prolonged physical inactivity being considered a major risk factor for human health and a reduced life expectancy [12]. Sedentary behaviour has been characterised by reduced energy expenditure and was recognised even among individuals engaged in MVPA [13]. The World Health Organisation recommendations on the appropriate level of activity are only met in one of four adults [14]. Evidence from wearable devices monitoring the level of human physical activity reported a massive increment in sedentary-spent time [15,16], with increased exposure to risks in all demographics and age groups [17]. Individuals’ sedentary behaviour can be identified by using self-reported questionnaires on daily time spent in any sedentary activities and mainly in a sitting position [18]. 

Increased sedentary time (ST) has been associated with an increased risk for type 2 diabetes mellitus, metabolic syndrome [19,20] and cardiovascular diseases (CVD) [19,21]. When comparing the acute effects of LIPA with inactivity in various populations, an impact on the cardiometabolic system [22], haemostasis [23], glucose and insulin responses have been observed in the case of inactivity [24]. When the time of sedentary behaviour, LIPA time and MVPA time were objectively analysed, exposure to risks was revealed in the case of sedentary behaviour in various age groups [25]. Physical inactivity and mainly prolonged sitting are involved in the mechanisms regulating proteins involved in disease susceptibility [26]. Due to daily reduced levels of muscle contractions, while maintaining a prolonged sitting position, modern society is experiencing the negative impact of inactivity [27]. Short-term studies demonstrated the unhealthy potential of one day of inactivity [28], with acute physiologically secondary effects of sitting being shown even in the active population [29]. Simulated microgravity has been shown to induce marked lower limb skeletal muscle atrophy when one limb was suspended for four weeks, and a similarity in the magnitude of muscle mass and strength reduction was obtained in the case of bed-rest [30]. 

Six weeks of unilateral suspension of the lower limb showed changes in the muscle morphology [31]. 

Few studies analysed the impact of sedentary behaviour on the muscles acting around the ankle joint, and even less data is available for healthy individuals. 

In patients suffering from diabetes, prolonged sitting revealed a potential negative impact on the foot plantar skin health [32]. In subjects that underwent ankle immobilisation, no significant differences were seen at 48 h, but significant differences were seen at one week. This presumed that the accumulated effects of immobilisation are needed to explain the reduction in strength at one week [33]. When participants were placed under chronic unloading (90 days of simulated microgravitational situation), negative effects resulted on the mechanical properties of the human Achille’s Tendon, while when participants were placed under resistive exercises, preventive effects were observed [34]. The effects of unloading on the tendon’s mechanical properties when placed for six weeks on bed rest conditions were also reported [35]. 

Despite sitting being associated with physical inactivity and inactivity is further associated with risks, the amount of sitting time linked with risks for human health has not yet been defined [36,37]. Reeves et al. suggested that in the case of simulated microgravity, in order to completely prevent alterations in the Achille’s Tendon mechanical properties, a certain level of muscle exercise is required [34].

Reducing sedentary behaviours was already strongly recommended and might be considered a preventive strategy [7]. 

One of the multiple proposed strategies for reducing sedentary behaviour was increasing the number of daily steps; thereafter, foot abilities to generate strength and endurance are requested for an effective gait. Walking is efficient if supported by the lower limb’s performance. Sitting, mainly prolonged sitting, as well as the association of sitting with sedentary behaviour, places one’s feet in an inactivity situation. Assessing foot muscles performance, despite being a difficult procedure that implies both technologies and testators’ skills, is essential. Determining the relationship between muscle action dose–response in particular functional activities could explain the side-effects of muscle inactivity and, in particular, during being in sitting positions. 

No agreement has been stated on the most appropriate method for measuring the strength of the foot intrinsic muscles [38]. From electromyographic studies [39] to toe flexor muscles (TFM) custom-made dynamometry [40], diverse methods for measuring foot intrinsic muscle’s function have been described [34].

By capturing ankle torque through the evaluation of maximal voluntary isometric contraction (MVIC), foot and ankle muscles that generated strength during a particular time interval and at the selected range of ankle joint motion can be precisely measured. Custom-made electronic dynamometry showed to be a reliable method for the measurement of ankle torque in humans [41] and a reproducible dynamic method when foot muscle strength was assessed in two moments in time [42].

Acute and accumulated effects of sitting on the intrinsic foot muscles and all muscles acting around the ankle joint need in-depth research, and the capturing of ankle torque by dynamometric means could be of relevance in both clinical and experimental fields. 

This study is, to the best of our knowledge, the first to examine the impact of active and sedentary behaviour on ankle torque when assessed with a custom-made electronic dynamometer. 

The aims of this study were: to analyse the impact and compare the effects of two different types of activities (short-time sedentary versus a short-time active behaviour) on the evolution of peak ankle torque in time; to assess the two types of routine lifestyles (sedentary and active) on ankle torque when participants were subjected to a short-time active and a short-time sedentary behaviour. 

Despite the impact of sedentary behaviour on ankle torque as the main focus of this paper, describing the measurement system (custom-made electronic dynamometer) as an innovative way to assess muscle strength in relation to inactivity is of great importance. In order to better profit from the measurement device’s practical use, we considered that an extensive outline of the measurement principles and measurement system description is required to ensure that any further replication of our study would be conducted with ease when a custom-made device is being used. 

The more time either short-time static or short-time active behaviour was maintained, the less force was observed during ankle torque testation. Peak torque during maximal isometric contraction was higher during a short time spent in active behaviour. 

## 2. Materials and Methods

### 2.1. Participants

Eight healthy adult consenting participants were selected for the study, and written signed consent was obtained before the enrolment. Ethical approval from the University of Medicine and Pharmacy “Victor Babes” Timisoara Ethics Committee was released and registered under Nr. 50/21.09-14.10.2020. The included participants had their measurements of peak ankle torque captured at maximum 2-weeks intervals in October 2021. All measurements were performed in the same physiotherapy unit placed in Timisoara, Romania. Based on their daily average number of steps recovered from the wearable devices, four of the participants were considered routinely active (≥6000 steps/day), and the other four participants were considered routinely sedentary (<6000 steps/day), further named routinely active and sedentary group, respectively. Only the average number of daily steps of the last month were recovered from the participant’s smartwatches. The data extracted from the devices did not represent the exact type of activity the participants underwent during their last month. Thereafter, neither physical activity or physical fitness had been recognised or identified from the recovered data. Nor a systematical control of the data derived from smartwatches, nor was a correlation between the data and participant reports applied for a possible correlation with a specific type of activity during the last month. Routinely active participants self-declared an active lifestyle during their occupational work, including light to moderate physical activity level and a regularly active lifestyle during off-work hours, respectively. Three of the four routinely active participants were engaged in recreational sports during the week, declaring a history of participation in performance sports. One participant practised soccer for eight years, one participant was a performant swimmer for five years and one participant practised acrobatic dancing for six years. All four routinely sedentary participants declared a sedentary lifestyle during their occupational work, including mainly sitting posture-spent time and no specific light to moderate intensity of physical activity during off-work hours. Only one participant from the four routinely sedentary group declared a history of 12 years of professional gymnastics.

We considered for exclusion any systemic diseases affecting the foot and ankle, past/present foot or lower limb trauma or surgery interventions that might have altered the foot mobility or function, or physical congenital foot and ankle deformities or malformations. Any cognitive/neurological conditions altering lower limb functionality, as well as psychiatric issues affecting the participant’s ability to participate, were also considered exclusion criteria. 

### 2.2. Data Collection

Anthropometric data, including age, gender, height, weight and foot length, were registered/measured at the first visit.

The average number of daily steps during the last month prior to measurements (independent of being considered physical activity or physical fitness) were registered after being recovered from the participant’s wearable devices (smartwatches), and a general physical activity level was established for each enrolled participant [43].

For a better framing of the off-work hours type of activity level, questions on daily activities were asked, targeting overall sedentary time spent in a sitting position, screen and tv time [18].

### 2.3. Description of the Measurement System and Measurement Procedure 

#### 2.3.1. Measurement System Description 

Bilaterally ankle torque measurements were performed for all participants using a reliable [41], reproducible [42], portable custom-made electronic dynamometer [44]. A simple diagram with the measurement system components is represented in Figure 1. 

The dynamometers pedal construction permitted the measurement of ankle torque at different joint angles by setting the desired pedal angle using an electronic inclinometer. Torque was converted into force, permitting that the applied force on the load cell was to be further converted into voltage. By the connected load cell amplifier, the load cell imbalance was converted into voltage, further evaluated with an oscilloscope (PicoScope Model 2204A) [45] connected to the personal computer (PC). The PicoScope^®^6 software used (manufactured by Pico Technology, PC Oscilloscope software version: 6.14.54.6108Copyright © 1995–2021, Pico Technology Ltd., St Neots, UK) [46] permitted the data acquisition and the recording of the whole measurement period. The apparatus construction, calibration and the full disclosure of the measurement system and protocol intervention have been described elsewhere [41]. A representation of the measurement system components and the participant’s general position on the chair is captured in Figure 2. Figure 2 demonstrates the minimum required amount of space (~3 m^2^) needed for the whole measurement system components and the participants’ chair to be positioned in order to proceed to measurements. 

The PicoScope^®^6 software graphic user interface allowed for the particular parameter configuration seen in Figure 3. 

During active ankle plantar flexion, a positive voltage value variation was obtained, as seen in Figure 4—trace A, while during dorsiflexion, a negative voltage value variation was obtained, as seen in Figure 4—trace B. The voltage values were transformed into torque in Nm. The actual voltage value, represented in V, corresponds to the voltage signal seen on the PC screen. Two sources of displacement of the voltage off-set are present in the set-up: the pedal remanent off-set (own pedal weight and permanent mechanical tension), as seen in Figure 4—trace E, and the participant-generated off-set, as seen in Figure 4—traces C and D (comprised of the lower limb weight in the absence of any ankle motion without tension generated by the fixation belt, and the lower limb weight in the absence of any ankle motion with the tension generated by the fixation belt, respectively). 

After each measurement, the data recordings were saved as a folder from the PicoScope interface, containing 32 text files of voltage values. For the validation of each measurement, a MATLAB [47] application was developed. The application allowed for the inspection of the saved text files by loading the multi-file contents of each measurement and concatenating them in a single time graph. By inspection of the resulted time graphs the operator appreciated the quality of each measurement and concluded for validation. Time graphs not passing the validation procedure were followed by another measurement trial. Measurements considered valid received a numerical code for each participant. The saved valid measurements were later sent for data processing. For accurate time graphs, low-pass filtering and scaling with the pedal constant were applied [41]. Voltage offset and peak torque during MVIC were processor-estimated and summarised into an Excel spreadsheet [48]. The summarised Excel data of voltage were converted into torque data [41]. Both time graphs of the voltage data, as well as time graphs of torque data, were visualised as graphic representations in time of the voltage variations in V, as seen in Figure 5a, or torque variations in Nm, as seen in Figure 5b–d. Automatically computed peak torque values (highlighted with a red circle, as seen in Figure 5c,d) and offset means (indicated by the red line in the time graphs, as seen in Figure 5c,d) were included in the torque time graphs. 

#### 2.3.2. Methods of Participant Preparation for Measurement Procedure 

Each measurement session consisted of measuring ankle torque during plantar flexion and dorsiflexion using a custom-made electronic dynamometer. All measurements were performed in the same testing laboratory with a constant room temperature (22 °C), allowing for an acclimatisation period of one hour before measurements. All possible adverse effects and the complete measurement protocol [41] were fully detailed to the participants. Possible adverse reactions, such as muscle cramps/fatigue/pain, or any other physical/emotional discomfort, were written and verbally presented to the participants and followed by cessation of the session in case such reactions might appear during the measurements. The participants were encouraged to fully relax while maintaining a sitting position with their trunk resting on the chair back-rest. Knee and hip joints were kept flexed, and the examined foot was resting in a plantigrade position on the dynamometer plate. Using nonelastic fibre belts, the fixation of the foot and thigh were ensured. For accurate measurements, the thigh strap was fixed just above the knee joint using one rigid fixation belt, and the foot was fixed in place using a second rigid fixation belt placed just above the dorsal aspect of the metatarsal-phalangeal joints (MPJ). This specific fixation allowed for the foot and ankle to remain stable and the heel to remain fixed in place during all measurements, as shown in Figure 6a,b. The foot positioning considered the ankle joint axis of rotation (defined as the line passing through the ankle malleoli) being aligned with the dynamometer’s pivotal point. The dynamometer’s pivotal point is marked with a blue horizontal line, as represented in Figure 6b. 

A more flexed knee position [49] allowed for the better isolation of the foot’s small muscles (flexor digitorum longus, flexor hallucis longus and the long toe flexor muscles) also involved in plantar flexing of the ankle. The strap fixation just above the MPJs level allowed for the attenuation of the anterior tibialis muscle impact as an ankle dorsiflexor. 

After all of the participants understood the type of requested muscle efforts, a succession of contractions followed as an experimental trial for better participant acclimatisation. Voltage acquisitions started while the clinician vocally commanded the direction of the ankle movements. After the acclimatisation test, a trial of three consecutive MVICs were registered for each ankle while the participants actively plantarflexed and dorsiflexed the foot. The acquisition period for the three consecutive MVICs was settled for 32 s. To better prevent muscle fatigue, a two-minute recovery break was permitted between the measurements for all pedal inclinations and all muscle groups. We measured the passive moment at rest and ankle torque at 0°, +5°, −5° of pedal inclination during three consecutive MVIC of 5 s, each separated by 5 s relaxation time for both plantar and dorsiflexor muscle groups, resulting in 12 measurements for each participant. The selected pedal inclinations (0°, +5°, −5°) corresponded to the same range of the participant’s ankle joint. When the tibia’s long axis was perpendicular to the ground, having the foot resting on the dynamometer at 0°of pedal inclination, we defined the ankle as being in a neutral position (90° of ankle dorsiflexion). From the neutral ankle position, we considered 95° of ankle dorsiflexion when +5° of pedal inclination and 5° of ankle plantar flexion when −5° of pedal inclination. 

All participants were measured during two separate six-hour sessions. One session comprised six hours of sedentary behaviour (further named static behaviour) while maintaining a prolonged sitting position. The other session comprised of six hours of active behaviour (further named active behaviour) defined by low and moderate levels of physical activity. 

A two-week interval was allowed between the two separate sessions.

The static behaviour was represented by six hours of prolonged sitting posture, with only small breaks allowed for participant’s urgent personal needs. The active behaviour consisted of six hours of mixed activities (short or long-distance walking, ascending/descending stairs, orthostatic postures) without exceeding the moderate level of physical activity. 

Three ankle torque measurements were performed at three different moments during each individual session: at two, four and six hours. The first measurement on each session was performed after two hours of the initiation of static/active behaviour.

The ankle torque measurement recordings from both the first and second sessions were later used for data processing. 

#### 2.3.3. Methods for Validation of Acquired Data 

One validation procedure during the measurements and two validation procedures during the interpretation of the results were performed for all of the acquired data. Participant-related errors (due to indiscipline or improper clinician commands) appeared and were considered errors during the measurements. When such errors were encountered, new measurements were requested. 

All recorded voltage time graphs resulting from the oscilloscope were inspected by the main researcher immediately after each performed measurement, and only valid measurements were selected. Errors were kept for statistical analysis without being considered valid measurements. 

Participant errors derived from off-set instability, improper contraction/break time (s), muscle efforts not corresponding with the clinician’s vocal commands, insufficient number of MVIC, fatigue, insufficiently sustained MVIC, pain or participant’s errors derived from testator command are possibly seen during measurements, and some of the most commonly encountered errors are represented in Figure 7a–d. 

By a simple analysis of the time graphs obtained with the developed application, the human or apparatus errors were easily recognised. Improperly recorded data derived from both participant and/or operator errors or apparatus errors were eliminated.

The obtained results from the three consecutive MVIC, the difference between the maximum obtained level of torque and the minimum obtained level of torque (defined as peak torque in Nm), were registered and statistically analysed. 

### 2.4. Other Clinical and Functional Performed Tests

A hand grip strength (HGS) test [50,51] for both right and left hands was performed using a factory-calibrated Hand Grip Dynamometer KERN MAP 80 K1 [52]. The testation was in accordance with the manufacturer’s instructions. The participants were asked to stand in an upright position having their testing arm resting close to their trunk and the elbow flexed at 90°. Three consecutive maximal volitional hand grip contractions were requested using the dynamometer’s 80 kg spring, and the maximal value of strength in kg was registered for the right and left hands. Right/left arm dominance was asked and registered.

Bilateral calf circumference (CC) in cm at the calf’s greatest girth was registered after being measured using a flexible nonelastic ruler. To measure CC, the participants were asked to stand in an upright position having their feet apart at shoulder width with their body weight equally distributed on both legs [53]. 

The number of repetitions during the Calf Raise Senior Test (CRST) [54,55] was registered for all participants. The participants were asked, while barefoot with their knees in maximum extension, to raise their heels simultaneously as high as possible, maintaining the same range of movement for all repetitions. To ensure stability, keeping their fingers on a vertical wall was permitted. While CRST was performed, muscle fatigue and muscle pain appeared during testation, and the onset moment of both fatigue and pain were registered. 

The Calf Raise Test (CRT) [56] was performed starting from a unilateral weight bearing stance, and the number of repetitions for both legs was registered. Participants were asked to unilaterally raise their heel as high as possible, maintaining the same range of movement for all repetitions. To ensure stability, keeping fingers on a vertical wall was permitted. Muscle pain and muscle fatigue that appeared during CRT and the onset moment of both fatigue and pain were registered. 

The chair raise test [57] was performed, and the total number of repetitions of chair raises in one minute was registered. Participants were asked to repetitively fully stand up in an upright position having their legs apart with their body weight symmetrically distributed on both legs, and then to sit down on a chair and perform as many repetitions as possible. 

### 2.5. Statistical Analysis 

For the study group, demographic data were reported as the arithmetic mean ± standard deviation or with absolute and relative frequencies. The normal distribution was tested with Shapiro–Wilk’s test. 

The collected time graphs during measurements were analysed for errors. The peak torque was considered missing in case errors were found. Other missing data were due to fatigue, muscle pain or discomfort of the participant. 

Clinical measurements were compared between the moments of measurements (at 2, 4 and 6 h) and between the active versus static behaviour with one-way ANOVA for repeated measures, separately for each flexion, foot and degree. The *p*-value for the comparisons between the moments (different measurements in time) and the *p*-value for the comparisons between active and static behaviour was reported. The number of data entered in the one-way ANOVA with repeated measures was 8 for each moment (one measurement for each participant). 

When all the data were analysed (576 measurements), multivariate analysis was performed with linear mixed models for repeated data because the data were not independent between the participants (each participant had 12 measurements in each moment). Independent factors were considered, and they were entered into the analysis as fixed factors: the active/static behaviour, the moments, the flexion, the foot and the degree. The data were analysed and reported to the first considered moment at 2 h. The data at 0 h were not available due to the protocol design.

Arithmetic means of the clinical measurements were computed. These arithmetic means were compared between the moments of the measurements with *t*-test for paired samples for normally distributed data and with Wilcoxon signed-rank test for non-normally distributed data. Post-hoc analysis was performed using Bonferroni correction.

Correlations between two parameters were analysed by computing the Pearson and Spearman coefficients of correlation. 

The *p*-value was considered statistically significant for values smaller than 0.05. Analysis was performed using SPSS application (manufactured by IBM Corp., Armonk, NY, USA, 2017) [58].

## 3. Results

### 3.1. Participant’s Characteristics 

Our participant’s characteristics are described in Table 1. The average participants’ age was 35.88 years old, with a minimum age of 23 and a maximum age of 58 years; three participants (37.5%) were male; four participants (50%) reported an average daily number of steps higher than 6000; five participants (62.5%) had a normal body mass index (BMI). For the studied sample, the HGS, CC, CRST, CRT, chair raise test and muscle pain and fatigue moment of onset during CRST, CRT and chair raise test for all participants and results are reported in Table 1. 

### 3.2. Clinical Measurements Results 

In Table 2, the descriptive statistic parameters for peak ankle torque during MVIC (dMVIC) in the case of active and static behaviour for each moment in time were presented and analysed with an ANOVA repeated measure for each foot, flexion and degree. The number of data entered in the ANOVA repeated measure analysis was *n* = 8 (one measurement of dMVIC for each participant). 

When we analysed all the data with linear mixed models without taking into consideration the foot, the flexion or the degree, we found that the active versus static behaviour (*p* = 0.005) and the moments (2 h vs. 6 h *p* = 0.040, 2 h vs. 4 h *p* = 0.128) had a significant effect on the dMVIC. The number of data entered in the linear mixed models’ analysis was *n* = 288 (for each subject 36 measurements of dMVIC).

The averages of dMVIC measured on different flexion, foot and degree on different moments per static/active behaviour are presented in Figure 8. There were significant differences between dMVIC after two hours compared with dMVIC after six hours (*p* = 0.019). There were no significant differences between dMVIC after two hours compared with dMVIC after four hours (*p* = 0.224) and also between dMVIC after four hours compared with dMVIC after six hours (*p* = 0.815). The average dMVIC in the case of active behaviour was significant statistically greater than the average dMVIC during static behaviour. 

When we tested the other factors: age, gender, BMI, average steps, AHGS, ACC, CRST, CRT, chair raise test, muscle pain during CRST, muscle pain during CRT, muscle fatigue during CRST and muscle fatigue during CRT in a repeated-measure ANOVA model, we did not find any significant influence on average dMVIC. 

We analysed the impact of the daily activity as the average number of steps (Table 3) or as the difference between the group of routinely active and routinely sedentary, but we did not find any significant statistical correlation (r = −0.313, *p* = 0.450) or association (*p* = 0.882) with dMVIC.

When we tested the correlation with The Pearson and Spearman coefficients of correlation, the same not-significant statistical relationship between average dMVIC and the other factors was found (Table 3).

## 4. Discussion

Based on previous studies, we hypothesised that a sitting posture could negatively influence foot and ankle muscle strength. 

One day of inactivity was associated with an unhealthy potential on human health [28], and sitting showed secondary negative effects even in the active population [29]. 

In our study, both routinely active and routinely sedentary participants showed a significant statistically decrement in average peak ankle torque when subjected to six hours of prolonged sitting. In our group, foot and ankle muscle strength suffered a reduction over time even when participants were subjected to six hours of low to moderate physical activity.

Chronic unloading of negative effects on the Achilles’ tendon was demonstrated, but when resistive exercises were added, some preventive effects were observed [34]. 

Considering the results demonstrated by Reeves et al. [34], we could estimate that when some type of exercise is associated, preventive effects of inactivity might be installed. 

In our study, we demonstrated that all eight participants, when subjected to low to moderate physical activity, showed higher values for average peak ankle torque when compared to the values obtained when subjected to sedentary behaviour. 

In the case of traumatic events, immobilisation effects have been reported. A decrease in ankle plantar flexor muscle strength has been observed after one week of immobilisation, but no significant differences were seen after two days of ankle immobilisation. The accumulated effects of immobilisation in time could explain the differences [33]. 

In our study, we evaluated the acute effects of six hours of inactivity on the plantar flexors’ muscles through the measurement of ankle torque in nontraumatic events. There were significant differences between dMVIC after two hours compared with dMVIC after six hours (*p* = 0.019). We should further consider that the same study could be replicated related to traumatic events.

In our study, only after six hours of prolonged sitting a significant decrement in average peak torque was observed, with no significant differences between measurements at two hours and at four hours. We can state that in our group of participants placed under static behaviour, ankle torque decreased after the accumulated effects of prolonged sitting. 

In patients suffering from diabetes, some electrophysiological reports analysed the soleus muscle activity in different sedentary postures, comparing chair-sitting with squat-like sitting [32]. 

In our group of participants, when sitting activity (static behaviour) was compared with a low to moderate physical activity (active behaviour), we found that even in healthy individuals’, inactivity has negative potential on foot and ankle muscles strength.

Optimisation of the effects of muscle activity for promoting health in the general population and reducing sedentary behaviour strategies, daily contraction duration of skeletal muscle and the role of contractile duration were studied [7]. 

Our study reports the effects on ankle torque when individuals were placed in a short-time sedentary behaviour. Due to the insignificant reduction in torque at four hours but the significant decrement in torque after six hours, new considerations should be evaluated for proper preventive strategies to reduce the negative effects of prolonged sitting on foot and ankle muscle strength. 

Based on our findings on the acute effects of six hours of sedentary behaviour on ankle torque, we might consider that while individuals adopt a prolonged sitting position, interposing activity-based breaks after at least four hours of inactivity might positively influence lower limb muscle performance. We only found a significant statistically decrement in ankle torque after six hours of inactivity, but no significant statistically decrement was found after two hours and four hours. These findings could help in establishing more rigorous prevention strategies for reducing sedentary behaviour effects of prolonged sitting in individuals. Such prevention strategies could address modifications of break time and break frequencies, time of work in sitting postures and routine behaviour modifications, as other previous studies showed [59]. 

In sports medicine and rehabilitation medicine, foot and ankle muscle strength assessments are essential, and one indicator of muscle performance tested in clinical practice is measuring ankle torque by assessing MVIC [60]. 

As no agreement has been stated on the most appropriate method for measuring the strength of the foot’s intrinsic muscles [38], our study analysed peak ankle torque by measuring MVIC using a reliable [41] and reproducible [42] custom-made electronic dynamometer. 

As per our knowledge, no particular study has assessed the impact of sedentary behaviour on ankle torque using a custom-made electronic dynamometer; therefore, describing the measurement system in detail as an innovative way to measure muscle strength in relation to inactivity was of great importance. 

The hand grip strength cut-off values and CC have been stated in relation to muscle strength reduction in sarcopenia [50,53], and some correlations with lower limb muscle strength have been reported [51]. 

We found no significant statistically relationship between average dMVIC and the average values of HGS and CC, probably due to the small sample (8 participants). 

The number of repetitions during the Calf Raise Senior Test (CRST) [54,55] has previously been used to assess plantar flexor muscle strength.

We applied CRST to our group of participants, and muscle pain/muscle fatigue appearance and the onset moment of both fatigue and pain were registered. We found no significant statistical relationship between average dMVIC and the average values of CRST fatigue and pain during CRST.

The Calf Raise Test (CRT) for evaluating calf muscle properties is a well-known clinical method [56], and based on a systematic review, the standards have been reported.

When we tested CRT in our group, muscle pain and muscle fatigue appeared, and the onset moment of both fatigue and pain were registered. We found no significant statistical relationship between average dMVIC and the average values of CRT, fatigue and pain during CRT.

The chair raise test, another commonly used method for assessing lower limb muscle parameters [57], was performed on our group of participants, and the total number of repetitions of chair raises in one minute was registered, with no significant statistical relationship between average dMVIC and the average values of the number of repetitions during the chair raise test. 

Although a massive increment in sedentary-spent time was reported [16], with increased exposure to risks in all demographics and age groups [17], no particular study has analysed the impact of short-time sedentary time on ankle torque.

We found no significant statistical relationship between average dMVIC and the other factors, such as age, BMI, foot length and average no. of daily steps.

We analysed the impact of the type of daily activity (defined by the average number of steps, which differentiates the two groups of routinely active and routinely sedentary participants), but we did not find any significant statistical correlation (r = −0.313, *p* = 0.450) association (*p* = 0.882) with dMVIC.

Despite sitting being associated with physical inactivity and further health risks, the amount of sitting time linked with risks for human health has not yet been defined [36,37]. 

Understanding the acute effects of prolonged sitting on ankle torque could better frame the long-term effects of sedentary behaviour on both routinely active and routinely sedentary individuals.

Correlations between physical inactivity and diseases need a better understanding.

In this study, we took into consideration only the measurements at two hours from the initiation of the behaviour, but not measurements at the baseline. The authors think that they ensured that after maintaining two hours of static/active behaviour, all participants had the same level of physical activity before the initiation of measurements. However, a better approach can be considered. Ideally, the study should have implemented a baseline measure in the protocol with a levelled physical activity for each participant. We considered our study limitations to have not established a levelled baseline of physical activity for the participants before being subjected to the measurement in both types of behaviour. To consider the baseline measurements when reporting the impact of the testing moments and the impact of two types of behaviour on ankle torque would have been desirable.

We took into consideration only the measurements at two hours from the initiation of the behaviour. This further ensured that after maintaining two hours of static/active behaviour, all participants had the same level state of physical activity before the measurements. 

This particular limitation is also due to the inconsistency of the participant’s data recovered from their wearable devices (smartwatches). We could only recover an average of daily steps from the participant’s last month of activity. Unfortunately, the data extracted from such devices does not represent the exact type of undertaken activity, with physical activity nor physical fitness being identifiable from the recovered data. To better identify the exact type of activity through the data derived from smartwatches, a more systematic control should be considered and eventually correlated with participants’ reports of activity questionnaires/scales [61].

Another limitation of our study is due to the small sample. Because of that, the comparison between samples with eight data (Table 2) was not found to be statistically significantly different, but the same data, when compared with multivariate techniques, were statistically significant. 

Further studies should implement the baseline measurement in the protocol when a levelled physical activity before the measurements are ensured and identify the exact type of physical activity through modern technology or by using validated questionnaires. 

## 5. Conclusions

The more time participants maintained either short-term static or short-term active behaviour, the lower the average peak ankle torque resulted in both situations. Both routinely active and routinely sedentary participants showed a decrement of force in time when maintaining both types of behaviours, with the sitting position being associated with a lower value of average peak ankle torque during maximal voluntary isometric contraction. 

Future studies should target the establishment of a threshold for the time spent in a sitting position, sedentary behaviour, in relation to foot muscle strength and establish whether breaking the routine during a specific activity might positively change the muscle force results.

Our force measurement results could complete ergonomic improvements for the achievement of healthy foot status in individuals spending prolonged time in a sitting position and especially when sitting while working.

Future studies should consider repeating the experiment in other types of groups of participants and possibly in groups affected by different conditions. 

## Figures and Tables

**Figure 1 jcm-11-02474-f001:**
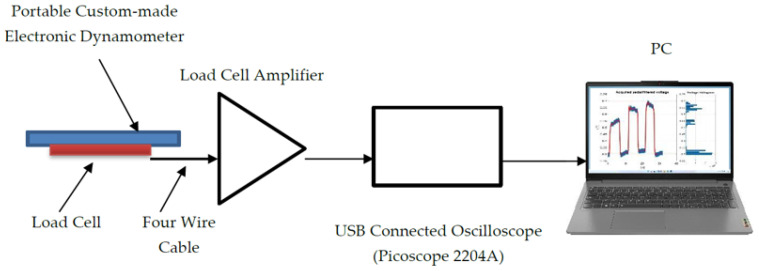
Diagram of graphic representation of the measurement system components comprising the portable custom-made electronic dynamometer, dynamometer load cell, a load cell amplifier connected to the dynamometer’s load cell through a four-wire cable, the USB connected oscilloscope (PicoScope 2204A) and the personal computer (PC).

**Figure 2 jcm-11-02474-f002:**
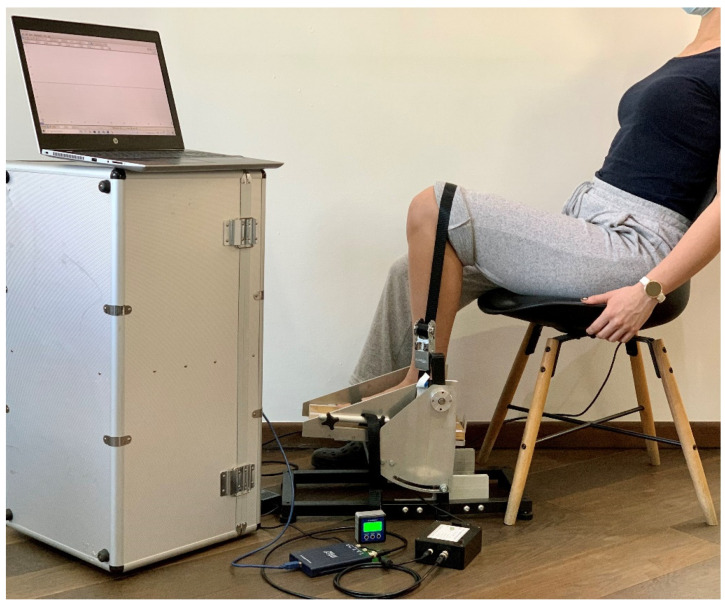
The system components used for ankle torque measurement and participant position on the chair; the measurement system components include: portable custom-made electronic dynamometer with an incorporated load cell, electronic inclinometer, a load cell amplifier, oscilloscope, connection wires and a PC.

**Figure 3 jcm-11-02474-f003:**
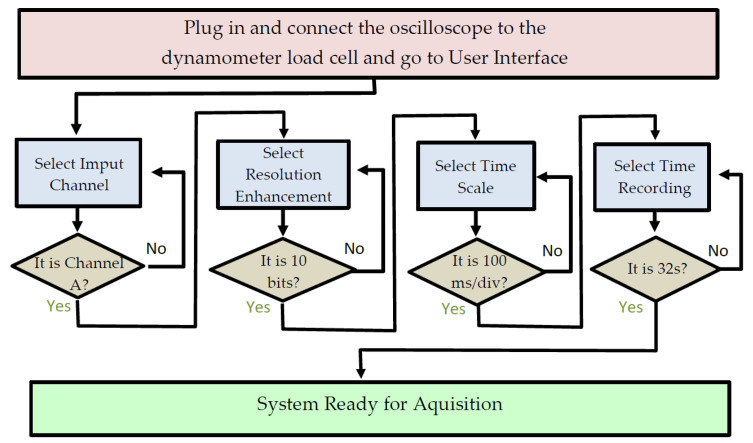
Flow chart representing the steps needed for the selection of the oscilloscope software graphic user interface configuration parameters: Channel A on, direct current (DC) coupling, input = 2 V/div, time base = 100 ms/div (32 s length record) and 10 bits resolution enhancement.

**Figure 4 jcm-11-02474-f004:**
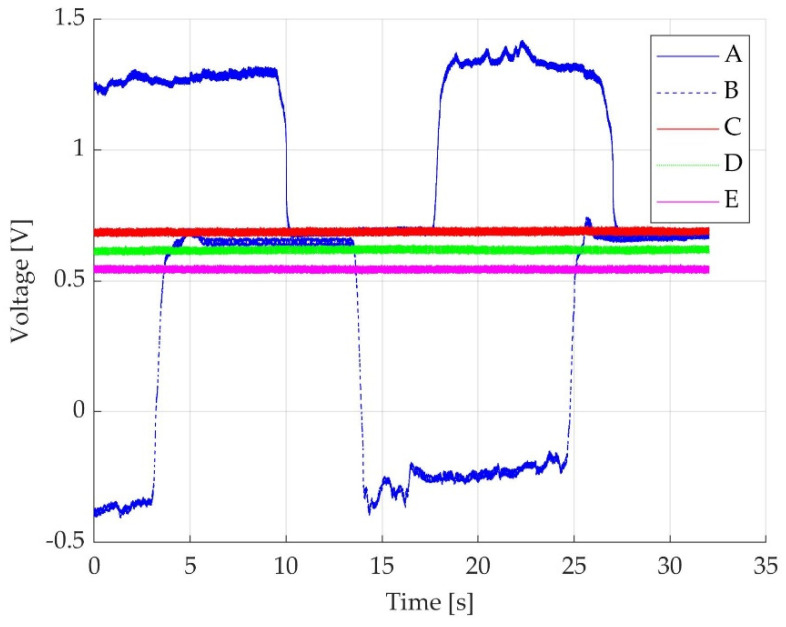
Time graph representing five different situations. Trace A represents a positive transition of voltage from the off-set level during two MVICs while the participant actively plantarflexes the foot; Trace B represents a negative transition of voltage from the off-set level during two MVICs while the participant actively dorsiflexes the foot; Trace C represents the off-set voltage level while the participant’s foot is relaxed on the pedal with fixation strap not tightened; Trace D represents the off-set voltage level while the participant’s foot is relaxed on the pedal with the fixation strap being tightened on the thigh just above the knee level; Trace E represents the pedal off-set in the absence of the participant’s foot on the dynamometer pedal.

**Figure 5 jcm-11-02474-f005:**
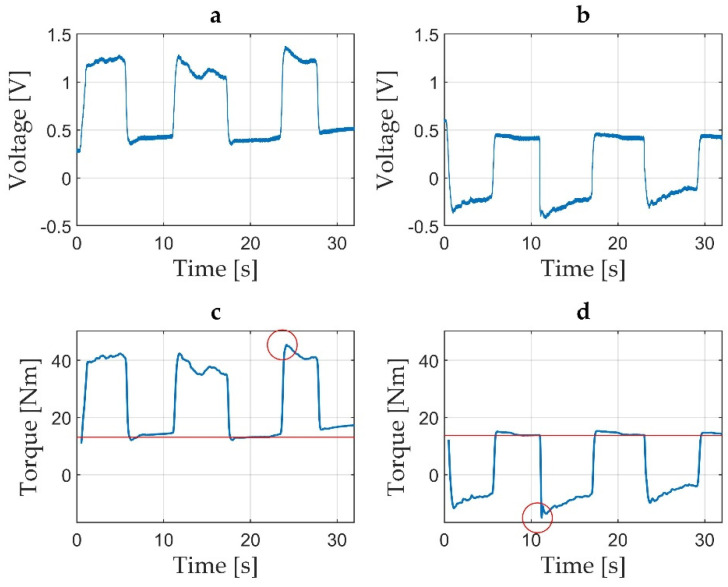
Three time graphs representing a succession of three MVIC of 5 s each followed by 5 s of relaxation between contractions: (**a**) time graph of voltage during ankle plantar flexion with the period being represented in seconds (s) on the Ox axis and voltage represented in V on the Oy axis; (**b**) an example of the same time graph for ankle dorsiflexion with the period being represented in seconds (s) on the Ox axis and torque represented in Nm on the Oy axis; (**c**) torque represented in Nm during ankle plantar flexion, where the peak torque is highlighted with a red circle and the mean off-set level is marked with a red line; (**d**) torque represented in Nm during ankle dorsiflexion, where the peak torque is highlighted with a red circle and the mean off-set level is marked with a red line.

**Figure 6 jcm-11-02474-f006:**
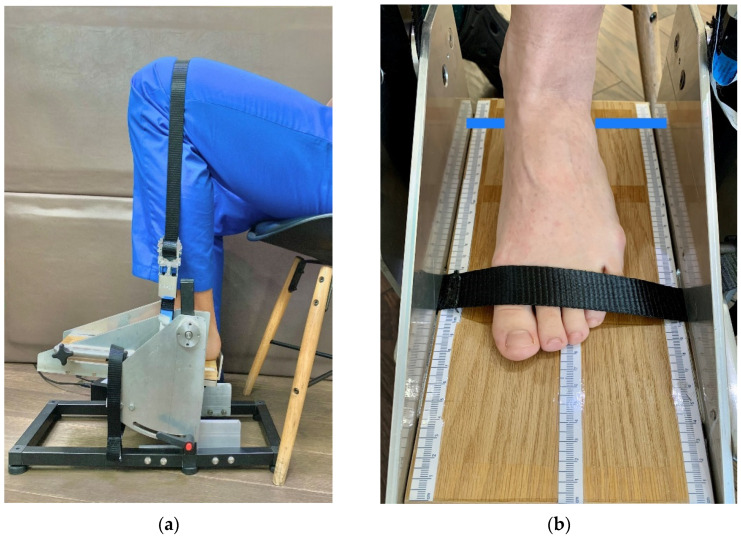
The participant’s sitting position: (**a**) fixation of the lower limb using a rigid fixation belt over the thigh right above the knee joint and the knee joint angle settled between 90° and 110°; (**b**) foot positioned on the dynamometer plate, with the ankle joint axis of rotation above the apparatus’s pivotal line marked on the device pedal with a blue line, with fixation of the rigid belt on the dorsum of the foot just above the MPJs level.

**Figure 7 jcm-11-02474-f007:**
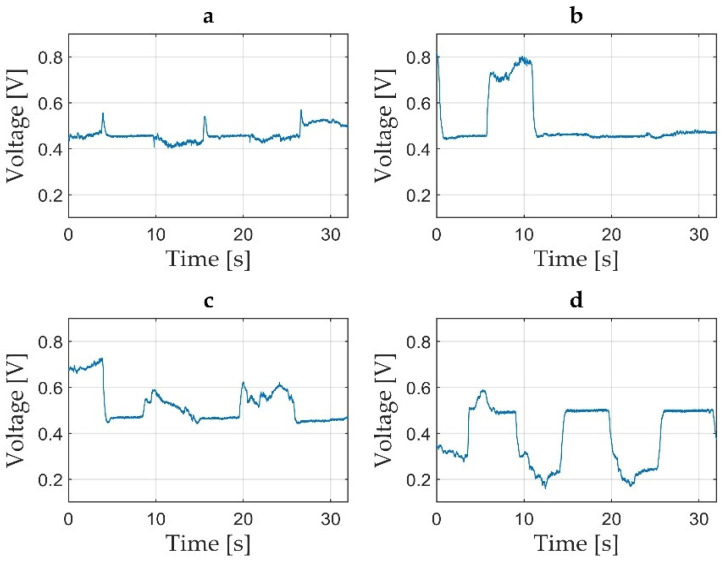
Time graphs representing invalid measurements due to errors derived from: (**a**) probable momentary fraction without participant’s proper sustain of MVIC; (**b**) insufficient number of MVIC probable due to the testator’s command error; (**c**) two insufficient and inconsistent MVIC following a good MVIC due to subject’s inability to focus on the testator command; (**d**) erroneous plantarflexion during dorsiflexion requests.

**Figure 8 jcm-11-02474-f008:**
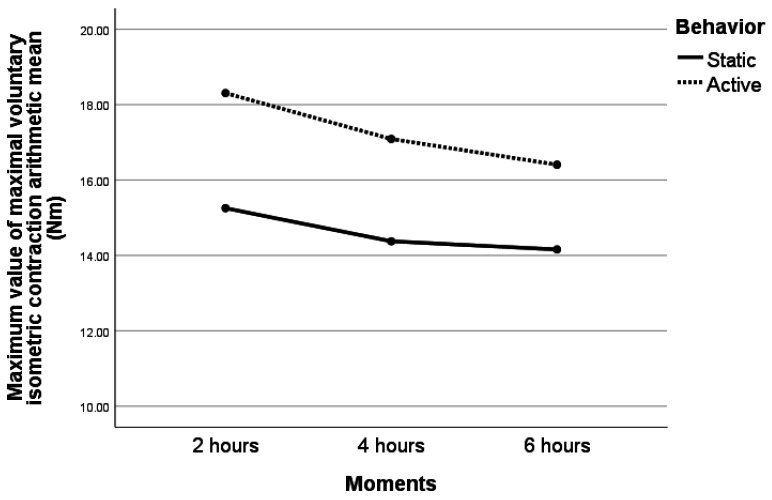
Impact of the moments of testing and the behaviour on dMVIC.

**Table 1 jcm-11-02474-t001:** Demographic and anthropometric data of the sample.

Parameters	Arithmetic Mean ± Standard Deviation (*n* = 8) ^1^
Age (years)	35.88 ± 12.65
Male, no. ^7^ (%)	3 (37.5)
Foot length (cm)	25.36 ± 2.56
BMI ^2^ (kg/m^2^)	25.02 ± 4.43
Average no. ^7^ of daily steps (steps/day)	6125 ± 2279.57
AHGS ^3^ (kg)	28.01 ± 10.62
ACC ^4^ (cm)	37.25 ± 3.17
CRST ^5^ (no. ^7^ of repetitions)	57.25 ± 17.69
Muscle pain during CRST ^5^ (no. ^7^ of repetitions)	42.75 ± 21.37
Muscle fatigue during CRST ^5^ (no. ^7^ of repetitions)	40.5 ± 10.57
ACRT ^6^ (no. ^7^ of repetitions)	30.44 ± 4.92
Chair raise test bilateral	41.88 ± 13.44
Muscle pain ACRT ^6^ (no. ^7^ of repetitions)	23.38 ± 5.71
Muscle fatigue ACRT ^6^ (no. ^7^ of repetitions)	25.31 ± 3.95

^1^ Total sample (*n* = 8); ^2^ BMI—Body Mass Index; ^3^ AHGS—Average Hand Grip Strength; ^4^ ACC—Average Calf Circumference; ^5^ CRST—Calf Raise Senior Test; ^6^ ACRT— Average Calf Raise Test; ^7^ no.—number.

**Table 2 jcm-11-02474-t002:** Descriptive statistics (arithmetic mean +/− standard deviation; median (25th; 75th percentile)) for dMVIC in the cases of active and static behaviours.

Flexion	Foot	Degree	Time Moment	dMVIC * (Nm)		*p* ** between Moments	*p* ** between Active and Static
				Static (*n* = 8)	Active (*n* = 8)		
Dorsi	Left	−5	2	17.78 ± 12.21	21.97 ± 14.05	0.087	0.560
			4	18.01 ± 10.1	21.35 ± 10.99		
			6	16.89 ± 11.08	19.01 ± 10.39		
		0	2	19.10 ± 12.40	26.11 ± 14.10	0.172	0.459
			4	18.01 ± 9.83	19.7 ± 10.82		
			6	18.02 ± 13.15	18.91 ± 11.96		
		5	2	20.2 ± 8.44	21.57 ± 11.03	0.150	0.632
			4	17.67 ± 8.22	19.73 ± 9.88		
			6	18.30 ± 12.15	17.24 ± 11.47		
	Right	−5	2	17.72 ± 12.33	23.99 ± 13.29	0.469	0.269
			4	18.46 ± 10.71	21.16 ± 11.66		
			6	17.60 ± 11.63	22.27 ± 12.01		
		0	2	15.76 ± 10.17	21.43 ± 12.85	0.098	0.783
			4	14.49 ± 8.19	21.32 ± 12.37		
			6	13.53 ± 9.56	18.44 ± 10.41		
		5	2	18.53 ± 10.5	21.58 ± 13	0.324	0.381
			4	16.26 ± 11.22	21.48 ± 12.03		
			6	15.29 ± 10.43	21.45 ± 14.04		
Plantar	Left	−5	2	14.7 ± 7.5	19.41 ± 11.31	0.922	0.267
			4	15.1 ± 7.88	18.50 ± 9.82		
			6	16.17 ± 8.50	17.10 ± 11		
		0	2	2.63 ± 5.10	5.95 ± 7.91	0.535	0.227
			4	3.17 ± 5.72	3.26 ± 6.99		
			6	3.83 ± 7.06	3.55 ± 7.66		
		5	2	17.84 ± 4.93	21.19 ± 17.43	0.340	0.313
			4	15.51 ± 4.72	20.48 ± 13.28		
			6	20.32 ± 13.75	19.50 ± 12.81		
	Right	−5	2	18.15 ± 9.38	20.50 ± 13.48	0.821	0.682
			4	18.10 ± 8.26	18.59 ± 11.04		
			6	16.59 ± 9.91	19.33 ± 14.99		
		0	2	3.49 ± 5.61	2.05 ± 1.21	0.752	0.188
			4	3.52 ± 5.87	1.71 ± 2.53		
			6	2.43 ± 4.78	2.31 ± 2.76		
		5	2	21.17 ± 8.10	20.69 ± 13.70	0.308	0.356
			4	17.9 ± 6.37	20.57 ± 12.59		
			6	16.87 ± 6.47	20.36 ± 14.47		

* dMVIC—peak torque (maximum value of maximal voluntary isometric contraction with pedal off-set correction), ** *p*-value from ANOVA with repeated measure.

**Table 3 jcm-11-02474-t003:** Correlation between average dMVIC and the other parameters.

Parameters	Pearson/Spearman Coefficient of Correlation	*p*
Age (years)	0.229	0.586
BMI ^1^ (kg/m^2^)	0.580	0.132
Foot length (cm)	0.483	0.226
Average no. ^6^ of daily steps (steps/day)	−0.313 *	0.450
AHGS ^2^ (kg)	0.573	0.137
ACC ^3^ (cm)	0.359 *	0.382
CRST ^4^ (no. ^6^ of repetitions)	0.535	0.171
Muscle pain during CRST ^4^ (no. ^6^ of repetitions)	0.569	0.141
Muscle fatigue during CRST ^4^ (no. ^6^ of repetitions)	0.365	0.373
ACRT ^5^ (no. ^6^ of repetitions)	0.411	0.360
Chair raise test bilateral	0.048	0.911
Muscle pain ACRT ^5^ (no. ^6^ of repetitions)	−0.252 *	0.585
Muscle fatigue ACRT ^5^ (no. ^6^ of repetitions)	−0.131	0.779

^1^ BMI—Body Mass Index; ^2^ AHGS—Average Hand Grip Strength; ^3^ ACC—Average Calf Circumference; ^4^ CRST—Calf Raise Senior Test; ^5^ CRT—Calf Raise Test; ^6^ no.—number. * Spearman coefficient of correlation.

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
