# Peer review of "Acute Effects of Sedentary Behavior on Ankle Torque Assessed with a Custom-Made Electronic Dynamometer"

_jcm, 2022, doi:10.3390/jcm11092474_

Round 1

Reviewer 1 Report

The manuscript is written methodically, with adequate explanations and satisfactory use of references where appropriate.

The clarity of writing and the organization are quite good.

The technical terms are explained in detail and the topic of the paper is clear and understandable.

The presented methodology and the results are clearly communicated, with the necessary background for the readers included in the paper.

The review of the state-of-the-art is sufficient.

The novel contribution of the paper is highlighted, as well.

The conclusion section includes some discussion about the results obtained by this work and the previous works on the analysis of the same or similar data.

Author Response

Thank you very much for your appreciations and review report.

We hope that we improved the manuscript as per your expectations.

Thank you very much for reading our work.

All the best, 

The authors

Reviewer 2 Report

The authors present a paper on "Acute effects of sedentary behavior on ankle torque assessed with a custom-made electronic dynamometer". It examines the effects with a two-group observational study design. In addition, body mass index, handgrip strength calf circumstance, calf raise senior test, and calf raise test was obtained. The authors used a customized electronic dynamometer. The current study's findings suggest that peak torque for calf respectively foot muscles could display the relevance of active habits in everyday life and show the impact of a sedentary lifestyle. However, the reviewer has major concerns about publishing the study in its current form.

General comments:

  1. I think the introduction is pervasive. The authors should shorten a couple of monotonous and redundant passages and focus on the preparation of discussion of their findings in the discussion and conclusion section. For instance, the sentences in lines 71-74 (on the risk factors of sedentary behavior) and lines 82-83 are redundant to the reviewer.
  2. In my opinion, the methods section is very extensive. The procedure of torque measurement is very detailed. I think the authors have to decide if the paper focuses on the measurement (like the referenced papers, for instance, ref. 41) or if the practical use for describing the impact of sedentary behavior is the focus. If it is for practical use, the device's usability has to be better outlined in the introduction and discussion section.
  3. The authors do not present the limitations of their study. Therefore, please add a limitations section.
  4. The reviewer has concerns regarding the statistical tests reported in the methods and results of the paper. The study represents a typical design for implementing an ANOVA with repeated measurements, in the reviewer's opinion. The baseline measurements should be considered when reporting the impact of the testing moments and the behavior (see also figure 8). Please implement the baseline measure in the protocol. If it is not possible, the issues of not concerning the baseline should be outlined in the limitations section. This would be a major concern and must be considered when interpreting the results.

More comments in detail:

Introduction:

  1. Lines 51-53: The authors should better outline the differentiation between PA and PF. The sentence appears to be unstructured to the reader.
  2. Line 109: In my opinion, the word for exercises should be "resistance exercises".
  3. Lines 125: Just a typo. Please delete the hyphen in the word "relationship".
  4. Lines 134-138 and 146-149: These passages are redundant. Please try to shorten the introduction as mentioned in my general comments (1).

Method:

  1. Lines 161-165: The sentence could be shortened (see general comment 1).
  2. Lines 185-187: Was the data recovered from the wearable devices without gaps? Please report the procedure in more detail. In my opinion, the data of such devices does not represent the natural PA and PF if it is not controlled systematically.

Results:

  1. Table 2: Please report the test statistics for differences in the three-time points in table 2. In a detailed description of statistical tests, an ES like Cohen's d should be reported.
  2. Table 3: Please highlight if the correlation coefficient is based on the Pearson or Spearman calculation (see lines 569-570).

Discussion:

  1. Lines 501-503: Please check the grammar of the sentence.
  2. Lines 507-510: The results are not discussed with the previous findings.
  3. Lines 529-531: This statement seems to be very general. Please try to better connect your findings with statements on the impact of changing people's PA.
  4. Lines 574-585: In my opinion, this part better fits the conclusion.
  5. Line 586: Please add a limitations section (see general comment 3).

Author Response

Thank you for your pertinent comments, suggestions and appreciations.

We hope that we reached your expectations, by answering point to point to all your remarks.

All the best!

The Authors

Round 2

Reviewer 2 Report

I have no further comments.